

# The E2F family as potential biomarkers and therapeutic targets in colon cancer

Haibo Yao[1], Fang Lu[2] and Yanfei Shao[3]

[1] Department of Gastrointestinal and Pancreatic Surgery, Zhejiang Provincial People's Hospital (People's Hospital of Hangzhou Medical College, Key Laboratory of Gastroenterology of Zhejiang Province), Hangzhou, China

[2] Department of Neurology, Second People's Hospital of Yuhang District, Hangzhou, China

[3] Department of Pharmacy, Zhejiang Provincial People's Hospital (People's Hospital of Hangzhou Medical College), Hangzhou, China

## ABSTRACT

**Background**. The E2F family is a group of genes encoding a series of transcription factors in higher eukaryotes and participating in the regulation of cell cycle and DNA synthesis in mammals. This study was designed to investigate the role of E2F family in colon cancer.

**Methods**. In this study, the transcriptional levels of E2F1-8 in patients with colon cancer from GEPIA was examined. Meanwhile, the immunohistochemical data of the eight genes were also obtained in the The Human Protein Atlas website. Additionally, we re-identified the mRNA expression levels of these genes via real time PCR. Furthermore, the association between the levels of E2F family and stage plot as wells overall survival of patients with colon cancer were analyzed.

**Results**. We found that the mRNA and protein levels of E2F1, E2F2, E3F3, E2F5, E2F7 and E2F8 were significantly higher in colon cancer tissues than in normal colon tissues while the expression levels of E2F4 and E2F6 displayed no significant difference between colon cancer tissues and normal tissues. Additionally, E2F3, E2F4, E2F7 and E2F8 were significantly associated with the stages of colon cancer. The Kaplan-Meier Plotter showed that the high levels of E2F3 conferred a worse overall survival and disease free survival of patients with colon cancer. Also, high levels of E2F4 resulted in a worse overall survival.

**Conclusion**. Our study implied that E2F3, E2F4, E2F7 and E2F8 are potential targets of precision therapy for patients with colon cancer while E2F1, E2F2, E3F3, E2F5, E2F7 and E2F8 are potential biomarkers for the diagnosis of colon cancer.

## INTRODUCTION

The E2F family is a group of genes that encode a family of transcription factors in higher eukaryotes (*Trimarchi & Lees, 2002*), which are currently subdivided into two groups based on their functional characteristics: namely transcriptional activators including E2F1, E2F2, and E2F3a and transcriptional repressors including E2F3b, E2F4, E2F5, E2F6, E2F7 and E2F8 (*Kwon et al., 2017*). The E2F family was originally regarded as a cellular activity that is required for the early region 1A transforming protein of adenovirus to mediate the

Corresponding author
Yanfei Shao, syf2019509@163.com

transcriptional activation of the viral E2 promoter (*Shen et al., 2018*). Subsequent studies unveiled that E2F family also controls the transcription of cellular genes that are responsible for cell division (*Stead et al., 2002*). The expression pattern of E2F activators is abnormal in multiple human malignancies, such as ovarian cancer (*Reimer et al., 2010*), breast cancer (*Millour et al., 2011*), bladder cancer (*Santos et al., 2014*), prostate cancer (*Kaseb et al., 2007*), lung adenocarcinoma (*Chen et al., 2016*) and colon cancer (*Tazawa et al., 2007*).

Colon cancer is a common malignancy, which is currently ranked as the third most prevalent cancer and the third leading cause of cancer death in the United States (*Tazawa et al., 2007*). Despite that considerable advancements in diagnostic and treatment methods, the 5-year overall survival rate of colon cancer remains no more than 21% (*Bagaria et al., 2019*). Hence, novel biomarkers and potential therapeutic targets should be screened to enhance prognosis and individualized treatments in patients with colon cancer. Of the eight members in E2F family, many genes have been disclosed to participate in the development of colon cancer in vitro and in vivo. For instance, E2F1 could upregulate the expression of c-Myc and p14ARF, therefore inducing apoptosis in colon cancer cells (*Elliott et al., 2001*). E2F2 was also proved to act as a potential new molecular biomarker for colon carcinogenesis (*Nicolet et al., 2008*). But to the best of our knowledge, bioinformatics analysis has yet been applied to investigate the roles of E2F gene family in colon cancer. Based on the analyses of thousands of gene expression or variation in copy numbers published online, we analyzed and validated the expression and different E2F transcription factors in patients with colon cancer in detail to explore the expression patterns, potential functions, and distinct prognostic values of transcription factors in colon cancer.

## METHODS

### Ethics statement
Our study was conducted based on the principles expressed in the Declaration of Helsinki, and approved by the Academic Committee of Zhejiang Provincial People's Hospital (Ethical Application Ref: IR2019001002). The datasets mentioned in our study were retrieved from the free and publicly available database, indicating that it was confirmed that all written informed consent was acquired.

### Comparison of the hub genes expression level
GEPIA is a newly developed interactive web server for analyzing the RNA sequencing expression data including 9736 tumors and 8587 normal samples from the TCGA database and the GTEx projects in a standard processing manner (*Li, Li & Chen, 2018*). GEPIA provides customizable functions including tumor/normal differential expression analysis, profiling according to cancer types or pathological stages, patient survival analysis, similar gene detection, correlation analysis, and dimensionality reduction analysis. In present study, we mainly employed the boxplot to visualized the expression of E2F genes in colon cancer tissues and adjacent tissues. Additionally, GEPIA was also employed to provide the transcripts per million (TPM) of E2F genes to display their relative expression level.

## The overall survival (OS) and stage plot of E2F

Similarly, we used the GEPIA database to get the overall survival (OS) and stage plot information of E2F genes of 135 patients with high level of E2Fs and 135 patients with low levels of E2Fs. The log rank $P$ value and hazard ratio (HR) with 95% confidence intervals were showed on the plot. $P < 005$ was statistically significant.

## Immunohistochemistry staining

The Human Protein Atlas (HPA, https://www.proteinatlas.org/) is a Swedish-based program initiated in 2003 with the aim to map all human proteins in cells, tissues, and organs using the integration of various omics technologies, including antibody-based imaging, mass spectrometry-based proteomics, transcriptomics, and systems biology. By acquiring immunohistochemical data of patients with or without colon cancer based on HPA, we further verified the expression of E2F genes.

## Gene Ontology and KEGG Pathway Analysis of E2F genes

Gene Ontology (GO) analysis is a common framework which can annotate genes and gene products including functions of Cellular Components (CC), Biological Pathways (BP) and Molecular Function (MF). Kyoto Encyclopedia of Genes and Genomes (KEGG) contains a set of genomes and biological pathways related with disease and drugs online database, which essentially is a resource for systematic understanding of biological system and certain high-level genome functional information. The Database for Annotation, Visualization and Integrated Discovery (DAVID, http://david.ncifcrf.gov, version DAVID 6.7) is an online bioinformatics database. It has covered a great many biological data and relevant analysis tools, then provide tools for the biological function annotation information for plenty of genes or proteins. $P < 0.05$ was considered as the cut-off criterion with significant difference. We could visualize the key biological processes, molecular functions, cellular components and pathways of DEGs by using DAVID online database. And further the scatter plot was performed by ImageGP (http://www.ehbio.com/ImageGP/index.php/Home/Index/index.html) according to the results of GO and KEGG pathway.

## Quantitative Real-Time PCR (qPCR)

Thirty-two pairs of colon tissues and surrounding adjacent tissues were obtained from patients diagnosed with colon cancer in Zhejiang Provincial People's Hospital. Total RNA (1 μg) was isolated using the TRIzol (Invitrogen, Carlsbad, CA, USA) kit, the concentrations and purities of which were quantified using an ultraviolet spectrophotometer. After that, cDNA was generated from RNA via reverse-transcribing using Transcriptor First Strand cDNA Synthesis Kit (Roche, USA) and followed real-time PCR was administrated using LightCycler 480 SYBR Green Master Mix (Roche Diagnostics GmbH) (*Xiao et al., 2018*). The expression levels of E2F genes were normalized to GAPDH. Relative mRNA expression levels were analyzed by the $2^{-\Delta\Delta}$ cycle threshold (CT) method. The primer sequences are displayed in Table 1.

**Table 1  Primer sequences used in this study.**

| Gene | Species | Forward | Reverse |
| --- | --- | --- | --- |
| E2F1 | Human | AGCATGATCCGAGATGTGGAA | TGCTCGCACGATCGTAGCCCT |
| E2F2 | Human | ACGATGTCGATGCTAGCGTGG | CGTCGTACCCAACTGCTAGCT |
| E2F3 | Human | ACGTCGTAGCTGATGGGCAGT | CGGTGTACGTACCAAAACTG |
| E2F4 | Human | ACAAATGCATGGGTCCGTCGA | GACATGCCGCCTGGAGAAAC |
| E2F5 | Human | ACGTGGACTGGCCCAACTGCC | GACATGCCGCCTGGAGAAAC |
| E2F6 | Human | CGCGTAGCTACGCTACAGCTAC | ACGTGATCGTAGCTGATCGCC |
| E2F7 | Human | CACACACGTTAAACACCAACCT | CGTGTGGGGCACGTGGCAAC |
| E2F8 | Human | ACAAAGTGCGGTCACGTTTCAT | ACGATCGATGCTGATCGCGA |

## TCGA data and cBioPortal

The Cancer Genome Atlas possessed both pathological data and sequencing of 30 different cancers. The liver colon carcinoma (TCGA, Provisional) dataset involving data from 618 samples with pathology reports was chosen for comprehensive analyses of E2Fs using cBioPortal. The genomic profiles included mutations, putative copy-number alterations (CNA) from GISTIC, mRNA expression z-scores (RNASeq V2 RSEM) and protein expression $Z$-scores (RPPA).

## Statistical analysis

All values were reported as means $\pm$ SD. A paired, two-sided Student's t test was used to compare differences between two groups. Statistical significance was analyzed by SPSS 19.0 software. Differences were considered significant when $P < 0.05$.

## RESULTS

### Expression levels of E2F genes in patients with colon cancer

By analyzing 349 normal colon tissues and 275 colon cancer tissues in GEPIA online website, we found the mRNA expression levels of E2F1, E2F2, E2F3, E2F5, E2F7 and E2F8 were significantly higher in patients with colon cancer than normal control ($P < 0.05$) while E2F4 and E2F6 displayed no significant difference between colon cancer group and control group ($P > 0.05$) (Figs. 1A–1H). Meanwhile, the above results were further identified using Oncomine database Fig. S1).

### Transcripts per million (TPM) of E2F genes

Transcripts per million (TPM) serves as a measurement of the proportion of transcripts in the pool of RNA. In our study, the higher TPM levels of E2F1, E2F2, E2F3, E2F5, E2F7 and E2F8 in colon cancer group were observed ($P < 0.05$), suggesting that the above 6 genes possessed more transcripts in colon cancer tissues. Consistent with the mRNA levels of E2F4 and E2F6, no significant difference was found between the 2 groups ($P > 0.05$) (Fig. 2).

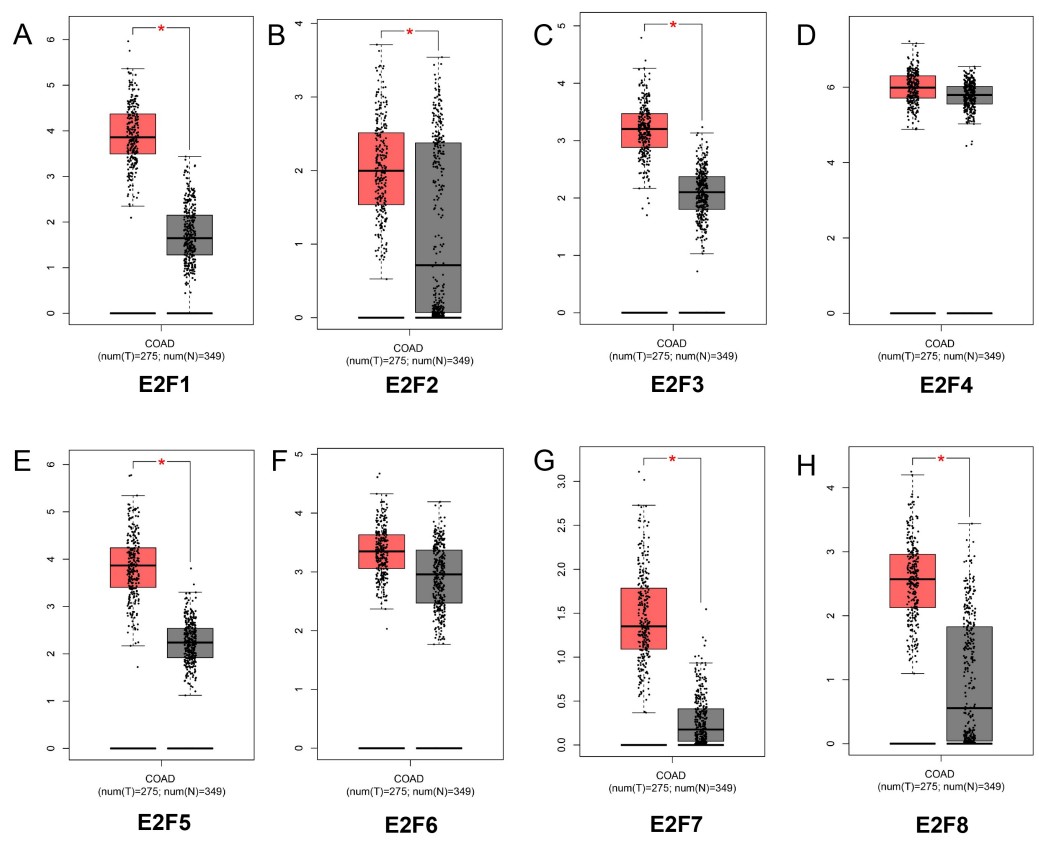

**Figure 1** The mRNA expression level of E2F1, E2F2, E2F3, E2F4, E2F5, E2F6, E2F7 and E2F8. (A) E2F1; (B) E2F2; (C) E2F3; (D) E2F4; (E) E2F5; (F) E2F6; (G) E2F7 and (H) E2F8.

## Correlation between E2F genes expression and tumor stage in patients with colon cancer

Subsequently, we analyzed the correlation between E2F genes expression and tumor stage in patients with colon cancer based on GEPIA online website. The results demonstrated that the expression levels of E2F3, E2F4, E2F7 and E2F8 displayed significant correlation with the tumor stage in patients with colon cancer while other members in E2F family in normal group and tumor group did not significantly differ (Figs. 3A–3H).

## GO and KEGG enrichment Analysis

The results (Table 2) from GO term enrichment analysis varied from expression levels and GO classification of the DEGs. By analyzing GO enrichment of these E2F genes via DAVID, we found that the E2F genes in BP were mainly enriched in transcription, DNA-templated, positive regulation of DNA endoreduplication, hepatocyte differentiation, negative regulation of cytokinesis and chorionic trophoblast cell differentiation. As for CC, the E2F genes were principally enriched in transcription factor complex. MF analysis uncovered that the E2F genes were mainly enriched in core promoter binding, transcription

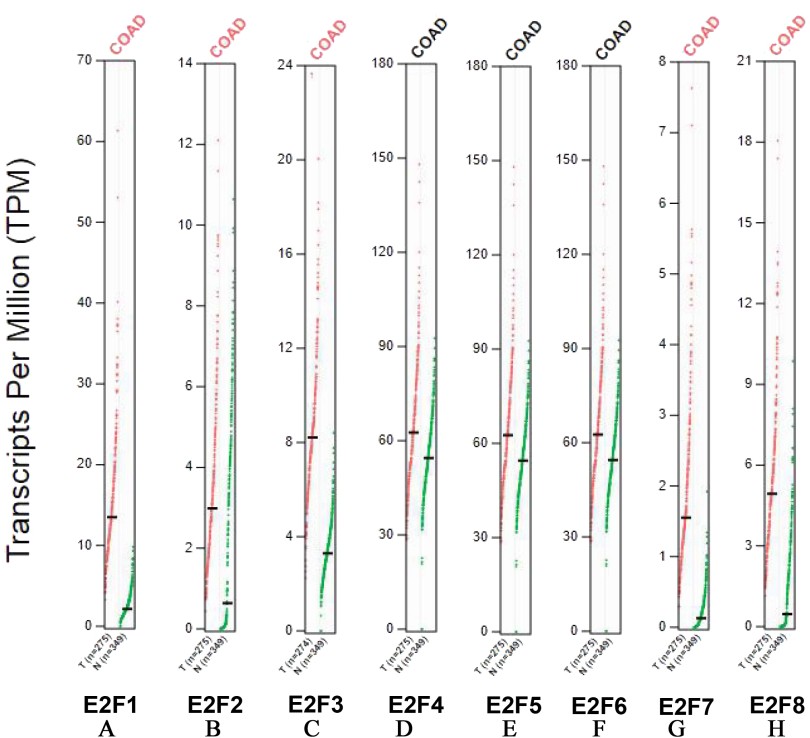

**Figure 2** **Transcripts per million (TPM) of E2F1, E2F2, E2F3, E2F4, E2F5, E2F6, E2F7 and E2F8.** (A)
E2F1; (B) E2F2; (C) E2F3; (D) E2F4; (E) E2F5; (F) E2F6; (G) E2F7 and (H) E2F8.

factor activity, sequence-specific DNA binding, DNA binding and transcription corepressor
activity.

To acquire a more comprehensive information regarding to the critical pathways of
those selected DEGs, KEGG pathways analysis were also carried out via DAVID. The results
in Table 3 unveiled the most vital KEGG pathways of the E2F genes, which were mainly
enriched in cell cycle and various malignant tumors including bladder cancer, non-small
cell lung cancer and pancreatic cancer.

## Correlation between E2F genes expression and overall survival in patients with colon cancer

Meanwhile, we further explored the potential association between the expression levels
of E2F genes and the overall survival of patients with colon cancer (Figs. 4A–4H). The
Kaplan–Meier showed that E2F3 and E2F4 displayed significantly correlation with the
overall survival of patients with colon cancer. To be more specific, the high levels of
E2F3 and E2F4 may contribute to worse prognosis of colon cancer ($P < 0.05$). Also, we
explored the disease free survival of these genes in patients with colon cancer. The results
(Fig. S2) showed that lower level of E2F3 may contribute to better disease free survival
in patients with colon cancer. Furthermore, we analyzed the E2F alterations by using the
cBioPortal online tool for colon cancer (http://www.cbioportal.org/). E2Fs were altered

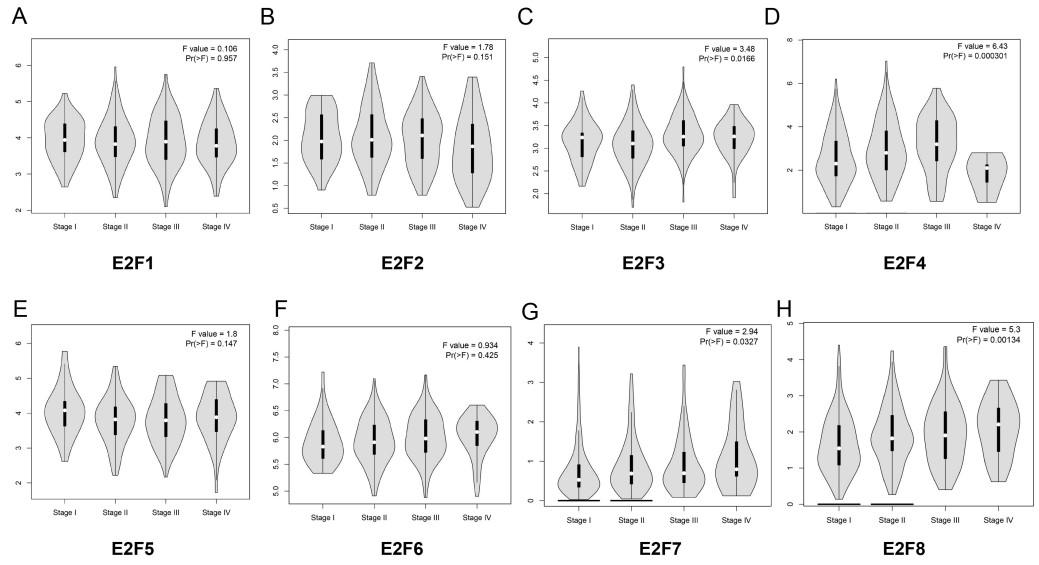

**Figure 3** **The relationship between the level of E2F genes and tumor stages in patients with colon cancer.** (A) E2F1; (B) E2F2; (C) E2F3; (D) E2F4; (E) E2F5; (F) E2F6; (G) E2F7 and (H) E2F8.

in 109 samples out of 618 patients with lung adenocarcinoma (17.7%). Two or more alterations were detected in almost one fifth of the samples (Fig. S3).

## The immunohistochemical staining of E2F genes

Additionally, we detected the protein expression levels of E2F genes based on the Human Protein Atlas online database. In accordance with the mRNA expression levels obtained from GEPIA, the protein expression of E2F1, E2F2, E2F3, E2F5, E2F7 and E2F8 were significantly higher in carcinoma tissues (Fig. 5).

## Re-identification of the mRNA expression levels of E2F genes

To further valid the expression patterns of E2F genes in colon cancer, we detected the mRNA levels of E2F genes in 32 patients with colon cancer using real time PCR. The results (Fig. 6) showed that the mRNA levels of E2F1, E2F2, E2F3, E2F5, E2F7 and E2F8 were significantly lower in adjacent tissues compared with those in carcinoma tissues (Fig. 6).

## DISCUSSION

The dysregulation of E2F genes has been reported in a great many cancers, including gastric cancer, lung cancer, hepatocellular carcinoma, pancreatic cancer and colon cancer (*Nevins, 2001*) (*Vacante, Borzì & Basile, 0000*; *Barbagallo et al., 2018*). Further bioinformatics analysis of colon cancer has to be further performed although the roles of E2F factors in the tumorigenesis and prognosis of certain cancers have been elaborated. To our knowledge, our study was the first time to investigate the mRNA and protein expression, and prognostic values of different E2F genes in patients with colon cancer. We sincerely hope our bioinformatics analysis could contribute to extra proof, optimize treatment

**Table 2   Gene ontology analysis of E2F family associated with colon cancer.**

| Category | Term | Count | % | *P*-Value | FDR | Genes |
|---|---|---|---|---|---|---|
| GOTERM_BP_DIRECT | GO:0006351~transcription, DNA-templated | 6 | 0.423429781 | 6.74E−09 | 5.53E−06 | E2F3, E2F4, E2F5, E2F6, E2F7, E2F8 |
| GOTERM_BP_DIRECT | GO:0032877~positive regulation of DNA endoreduplication | 2 | 0.14114326 | 8.11E−04 | 0.663544451 | E2F7, E2F8 |
| GOTERM_BP_DIRECT | GO:0070365~hepatocyte differentiation | 2 | 0.14114326 | 0.001622454 | 1.322792672 | E2F7, E2F8 |
| GOTERM_BP_DIRECT | GO:0032466~negative regulation of cytokinesis | 2 | 0.14114326 | 0.002027739 | 1.65081418 | E2F7, E2F8 |
| GOTERM_BP_DIRECT | GO:0060718~chorionic trophoblast cell differentiation | 2 | 0.14114326 | 0.002432892 | 1.977771789 | E2F7, E2F8 |
| GOTERM_CC_DIRECT | GO:0005667~transcription factor complex | 6 | 0.423429781 | 1.61E−10 | 8.27739543574068E−08 | E2F3, E2F4, E2F5, E2F6, E2F7, E2F8 |
| GOTERM_MF_DIRECT | GO:0001047~core promoter binding | 3 | 0.211714891 | 1.15E−04 | 0.068932147 | E2F3, E2F7, E2F8 |
| GOTERM_MF_DIRECT | GO:0003700~transcription factor activity, sequence-specific DNA binding | 4 | 0.282286521 | 4.73E−04 | 0.283673732 | E2F3, E2F4, E2F5, E2F6 |
| GOTERM_MF_DIRECT | GO:0003677DNA binding | 3 | 0.211714891 | 0.016757148 | 9.649556019 | E2F4, E2F5, E2F6 |
| GOTERM_MF_DIRECT | GO:0003714~transcription corepressor activity | 2 | 0.14114326 | 0.04355609 | 23.46393428 | E2F7, E2F8 |

**Table 3   KEGG pathway analysis of E2F family associated with colon cancer.**

| Category | Term | Count | % | *P*-Value | FDR | Genes |
|---|---|---|---|---|---|---|
| KEGG_PATHWAY | cfa04110:Cell cycle | 5 | 0.352858151 | 1.05E−07 | 7.14E−05 | E2F1, E2F2, E2F3, E2F4, E2F5 |
| KEGG_PATHWAY | cfa05219:Bladder cancer | 3 | 0.211714891 | 2.14E−04 | 0.145507524 | E2F1, E2F2, E2F3 |
| KEGG_PATHWAY | cfa05223:Non-small cell lung cancer | 3 | 0.211714891 | 4.16E−04 | 0.282144998 | E2F1, E2F2, E2F3 |
| KEGG_PATHWAY | cfa05212:Pancreatic cancer | 3 | 0.211714891 | 5.25E−04 | 0.355786534 | E2F1, E2F2, E2F3 |
| KEGG_PATHWAY | cfa05214:Glioma | 3 | 0.211714891 | 5.41E−04 | 0.366990838 | E2F1, E2F2, E2F3 |
| KEGG_PATHWAY | cfa05218:Melanoma | 3 | 0.211714891 | 6.46E−04 | 0.437793075 | E2F1, E2F2, E2F3 |
| KEGG_PATHWAY | cfa05220:Chronic myeloid leukemia | 3 | 0.211714891 | 6.64E−04 | 0.450188061 | E2F1, E2F2, E2F3 |
| KEGG_PATHWAY | cfa05222:Small cell lung cancer | 3 | 0.211714891 | 8.82E−04 | 0.597682584 | E2F1, E2F2, E2F3 |
| KEGG_PATHWAY | cfa05215:Prostate cancer | 3 | 0.211714891 | 9.69E−04 | 0.656358939 | E2F1, E2F2, E2F3 |

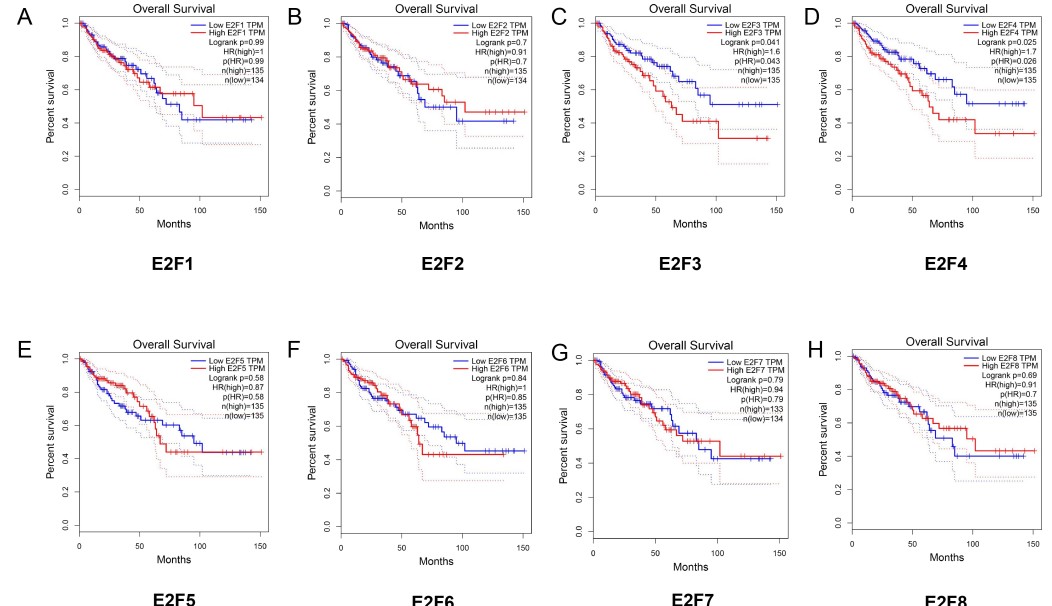

**Figure 4   The relationship between the level of E2F genes and overall survival in patients with colon cancer.** (A) E2F1; (B) E2F2; (C) E2F3; (D) E2F4; (E) E2F5; (F) E2F6; (G) E2F7 and (H) E2F8.

strategy, and improve the accuracy of diagnosis and prognosis for patients with colon cancer.

E2F1 is the most explored in colon cancer among the 8 genes of E2F family. E2F1 regulates G1/S-phase transition of cell cycle by transactivating multiple genes including chromosomal DNA replication and its own promoter (*Jiang et al., 2010*). E2F1 itself could be regulated in a cell cycle-dependent manner, mainly through temporal association with retinoblastoma, a pocket protein family member. In turn, pocket proteins could be regulated by phosphorylated cyclin-dependent kinase (*Vasavi et al., 2017*). The transcription of the thymidylate synthase gene was regulated by E2F1 in primary colon cancer specimens, the regulatory pattern of which from E2F1 to thymidylate synthase

   

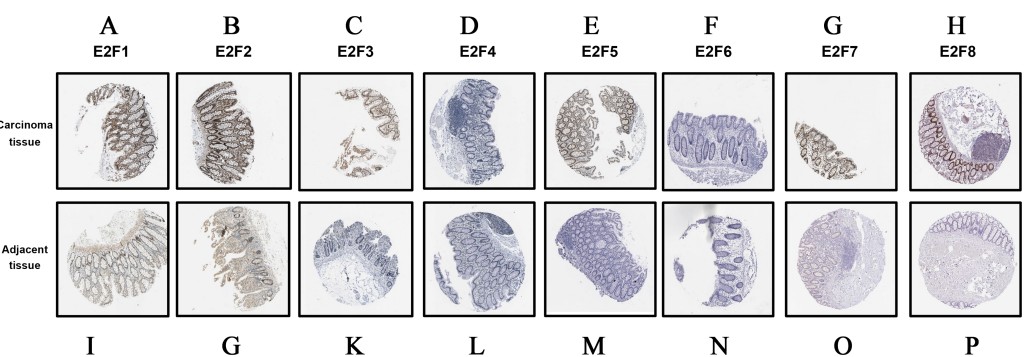

**Figure 5**  **The immunohistochemical staining data obtained from HPA database.** (A–H) The protein expression of E2f1, E2f2, E2f3, E2f4, E2f5, E2f6, E2f7 and E2f8 in carcinoma tissues. (I–P) The protein expression of E2f1, E2f2, E2f3, E2f4, E2f5, E2f6, E2f7 and E2f8 in adjacent tissues. Positive staining is brown while negative is purple. Image of the colon cancer is magnified 40 times under a light microscope.

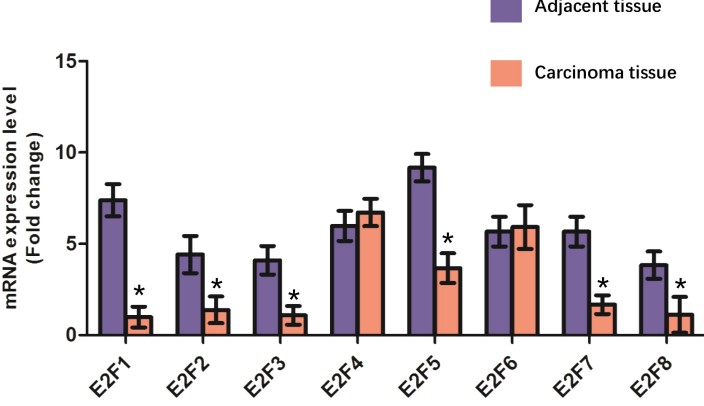

**Figure 6**  **Validation of E2F genes in the carcinoma tissues and adjacent tissues from patients with colon cancer using real time PCR.** * $P < 0.05$ vs carcinoma tissue.

may be highly conserved during malignant progression of colon cancer (*Kasahara et al., 2000*). Additionally, in p53 deficient human colon cancer cells, Mdm2 inhibition could trigger cell apoptosis by activating E2F1-and p73- mediated protein expression of Siva-1 and PUMA (*Ray, Bhattacharya & Johnson, 2011*). In our study, we found that the mRNA and protein expression of E2F1 was significantly higher in colon carcinoma, however, the expression level of which did not affect tumor stage and overall survival, suggesting that E2F1 may just serve as a tumor "switch" to induce tumorigenesis.

E2F2 plays dual roles in the occurrence and development of tumor. On the one hand, E2F2 can indeed serve as an "activator" to increase the expression of its targets (*Infante et al., 2008*). On the other hand, E2F2 can also be a suppressor by repressing cell cycle regulators to maintain quiescence and suppressing myc-induced proliferation and tumorigenesis (*Johnson & James, 2006*). For example, E2F2 could suppress the proliferation of T lymphocytes (*Azkargorta et al., 2010*). In fact, the expression level of E2F2 was very

low level in colon cancer and E2F2 inhibition could enhance proliferation and cell cycle of colon cancer cells by suppressing the expression of surviving and regulating the expression of C-MYC, CCNA2, MCM4 (*Li et al., 2015*) as well as CDK2 (*Xanthoulis & Tiniakos, 2013*). Our analysis unveiled that the mRNA and protein expression levels of E2F2 were significantly higher in colon cancer tissues. Similarly, the expression level of E2F2 was not associated with tumor stage and overall survival in patients with colon cancer.

E2F3 is critical for the transcriptional activation of various oncogenes controlling the rate of proliferation of both tumor and primary cells. For example, miR-125b could block the colonies form of bladder cancer cells in vitro and to develop tumors in nude mice by targeting E2F3 (*Huang et al., 2011*). In colon cancer, E2F3 serves as a direct target of miR-503 that is responsible for the proliferation and cell cycle distribution. The inhibition of E2F3 not only suppressed proliferation but also triggered apoptosis and G0/G1 arrest in SW480 colon cancer cells (*Huang et al., 2011*). In our report, the expression of E2F3 in colon cancer tissues was higher than that in normal tissues. We also disclosed that E2F2 expression was significantly correlated with tumor stage in patients with colon cancer. A high E2F2 expression was significantly correlated with poor survival in patients with LC.

E2F4 as well as its target cyclin A were significantly up-regulated and mostly nuclear in human colon tumor cells compared with the corresponding benign epithelium (*Garneau et al., 2010*). The mutations of E2F4 also enhanced the capacity of colon cancer cells to grow without anchorage, therefore giving rise to tumor progression (*Marie-Christine et al., 2013*). In our study, although there was no significant difference of the expression between colon cancer tissues and normal tissues, the level of E2F4 was associated with tumor stage as well as overall survival in patients with colon cancer.

E2F5 was a bona fide target gene of miR-34a and the restoration of E2F5 significantly antagonized the suppression of colon cancer cell proliferation and invasion (*Guifeng et al., 2015*). Our study demonstrated that the expression of E2F5 was significantly increased in colon cancer tissues, however, the level of which did not affect the tumor stage and survival rate of patients with colon cancer.

Up to now, there were no studies on the role of E2F6 in colon cancer published, which was consistent with our bioinformatic analysis, as evidenced by the undifferentiated expression levels of E2F6. Similarly, the expression levels of E2F6 had no effect on the tumor stage and overall survival of patients with colon cancer based on our findings.

The E2F7 gene, located at chromosome 12q21.2, contains 14 exons and acts as a tumor suppressor in the regulation of cell cycle progression (*Carvajal et al., 2012*). A significant association between E2F7 missense variant rs3829295 and CRC susceptibility, especially in males, was found in a Chinese Han population (*Ai et al., 2017*). Additionally, miR-520a could regulate inflammatory reactions by targeting E2F7 in colon cancer (*Cui et al., 2017*). Our findings disclosed that the expression of E2F3 had an association with colon cancer stage, meanwhile the expression of E2F7 in colon cancer tissues were significantly increased in patients with colon cancer.

As for E2F8, although there were no studies reporting its roles in colon cancer, our findings unveiled its different expressions between normal tissues and colon cancer tissues,

apart from its association with tumor stage. Hence, more experiments should be performed to explore the potential roles of E2F8 in colon cancer.

## CONCLUSION

Our studies indicated that the deregulation of E2F1, E2F2, E2F3, E2F5, E2F7 and E2F8 in colon cancer tissues might play a vital role in colon cancer oncogenesis, which could be promising diagnostic biomarkers for LUAD. In addition, the expression of E2F3, E2F4, E2F7 and E2F8 were significantly associated with tumor stages and overall survival of the patients with colon cancer, suggesting that they may serve as potential therapeutic targets for colon cancer. Altogether, we hope our study may be helpful to potential prognostic markers for the improvement of colon cancer survival and prognostic accuracy in the future.

### Funding
This study was supported by the Natural Science Foundation of Zhejiang Province (LQ19H160013). The funders had no role in study design, data collection and analysis, decision to publish, or preparation of the manuscript.

### Grant Disclosures
The following grant information was disclosed by the authors:
Natural Science Foundation of Zhejiang Province: LQ19H160013.

### Competing Interests
The authors declare there are no competing interests.

### Author Contributions
- Haibo Yao and Fang Lu conceived and designed the experiments, performed the experiments, analyzed the data, prepared figures and/or tables, authored or reviewed drafts of the paper, and approved the final draft.
- Yanfei Shao performed the experiments, analyzed the data, prepared figures and/or tables, authored or reviewed drafts of the paper, and approved the final draft.

### Human Ethics
The following information was supplied relating to ethical approvals (i.e., approving body and any reference numbers):

Our study was conducted based on the principles expressed in the Declaration of Helsinki, and approved by the Academic Committee of Zhejiang Provincial People's Hospital (Ethical Application Ref: IR2019001002). The datasets mentioned in our study were retrieved from the free and publicly available database, indicating that it was confirmed that all written informed consent was acquired.

## Data Availability

The raw measurements are available in the Supplemental Files.

## Supplemental Information

Supplemental information for this article can be found online at http://dx.doi.org/10.7717/peerj.8562#supplemental-information.

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
