# Peer review of "The E2F family as potential biomarkers and therapeutic targets in colon cancer"

_PeerJ, doi:10.7717/peerj.8562_

## Round 0.1 · original submission · Major Revisions

The reviewers had mixed opinions about your manuscript. While one reviewer appreciated the biomedical significance of the proposed study, two other reviewers had serious concerns about the biological motivation, novelty of the findings and rigor of the bioinformatics analysis (e.g., batch effects, enrichment analysis, etc).

The qPCR results seemed to come out from nowhere. There was virtually no description of the sample source! If they were from your own hospital, IRB approval is required. The data also needs to be deposited to a public database.

Please address reviwers' comments in the revision.

Reviewer 1 ·

Basic reporting

In this paper, authors utilize bioinformatics tools (such as GEPIA and DAVID) investigate the role of E2F family in colon cancer tool, followed by real time PCR validation. First, I don't know why author only investigate the correlation of E2F and colon cancer, rather than other cancer types? In addition, I don't think the method used in the study with enough novelty. The experiment description is not very clear as well. These are also lots of typo errors in their manuscript! Considering the current study provides limited insights, I do not feel that the paper overall presents a clear concept to warrant publication.

Some further comments are given below.

Q1. In the introduction part, why you only interesting in E2F and colon cancer, rather than other cancer types? The E2F genes show specific expression pattern in colon tissue (can check GTEx)? Is it more significant change pattern (fold change) between tumor and normal tissue? The author should explain this!

Q2: Line 70, about the sample number. In fact, there are only several hundred normal samples in the TCGA samples. The author should read the reference carefully~! The 8587 normal samples should come from GTEx database. You should add more clear description about your data used in the study!

Q3. Line 73, boxplot is just a method for visualization. It is not suitable to use detect

Q4. Section 2.3, add more description. How many groups? Where is the group information come from? I can not follow you if without Figure.

Q5. Section 2.4, please cite the reference of HPA.

Q6. Too many typo errors, e.g. line 87 Gene ontology (GO) should be Gene Ontology (GO). Similar to Cellular Component (CC) etc.

Q7. Line 92, about DAVID software, please cite the reference and provide the version! Line 96, similar to ImageGP software.

Q8. Line 99, 32 should be "Thirty-two". Arabic numerals can not be used as initial of one sentence.

Q9. About Section 2.5, please add more description for your method. What about the RNA quality for RT-PCR? RIN value > 7 or 8? How many technical for each gene? What about the sd? What software used? How about the correlation between RT-PCR and TCGA?

Q10. Section 3.1 and 3.2, I am confused, why you treat the expression level and TMD as two parts? Is there any difference? You just focus on relative compare? In addition, about the P value is raw p-value, or after multiple corrections? Please also specific which test you used. Check GEPIA.

Q11. Line 124, why you say strong correlation? Which test? P-value? Is there any statistical analysis?

Q12. Why you did enrichment analysis? There are only 8 genes! I think at least 50 genes are need for gene set enrichment analysis. If they are transcriptional factors, you should focus on their target genes!

Q13. Line 142, which test?

Q14. No conclusion part?

Q15. Figure 6, what's the error line? Standard deviation? I can't see the lower boundary.

Experimental design

no comment

Validity of the findings

no comment

Reviewer 2 ·

Basic reporting

This is a very interesting paper that investigated the role of E2F family in colon cancer.
In general the manuscript is very good and English language is adequate. Please check throughout the text for grammar and spelling errors; please provide a full explanation for all the acronyms when they are mentioned for the first time in the text (i.e. line 69 GEPIA).
The figures are very clear and useful.
I would suggest to include more statements in the discussion section and updated references in order to emphasize the growing importance of biomarkers for diagnosis and therapy of colorectal cancer (i.e. Vacante M et al, World J Clin Cases. 2018; Barbagallo C et al, Mol Ther Nucleic Acids. 2018).

Experimental design

Please in the statistical analysis paragraph, report the names of the tests used to calculate the statistical significance (i.e. t-test; chi-square).

Validity of the findings

The results look interesting and appealing. I would include a statement in the conclusion paragraph on the possible clinical or surgical implications of the results and future perspectives as regards the prognostic or predictive role of these novel biomarkers.

·

Basic reporting

No comment

Experimental design

No comment

Validity of the findings

Please add the references for real time PCR data.

Additional comments

This manuscript investigated the association between E2F family genes and colon cancers through a series of bioinformatics analysis on gene expression data, immunohistochemical data, and real time PCR data. Though the findings seem to be valuable, I am not fully convinced by some technical details in the manuscript. My comments are list as follow.

1. The main conclusion seems to have some conflicts with another E2F family study in colon cancer published on 2014 tilted as “The relationship between E2F family members and tumor growth in colorectal adenocarcinomas: A comparative immunohistochemical study of 100 cases”. It has already reported E2F1 and E2F2 as biomarker of colon cancer. So the sentence in the conclusion part of this manuscript: “E2F1, E2F2, E3F3, E2F5, E2F7 and E2F8 are novel biomarkers for the diagnosis of colon cancer.” needs to be revised.
2. A follow-up question of my question 1 is that what are the E2F family’s expression in tumor versus normal from TCGA COAD data only? TCGA COAD has around 40 normal samples which should be enough to reproduce or validate some results. My major concern is that GEPIA combines the normal samples from TCGA and GTEx. Although the batch effect correction is considered in GEPIA, it is still worth investigating the original TCGA tumor versus normal.
3. The significant p value cut-off is set to 0.05 which usually introduces inflated false discovery rate. It would be better to use p values after multiple testing correction.
4. In addition to overall survival analysis in figure 4. It is also recommended to use progression free interval (PFI). There are some reports that suggesting PFI in TCGA data has better follow-up comparing to overall survival.
5. What is the data source of real-time PCR? Please add the corresponding references.

---

## Round 0.2 · accepted · Accept

The reviewers are satisfied with the revision and recommend acceptance.

Reviewer 1 ·

Basic reporting

The author have addressed my concern.

Experimental design

no comment

Validity of the findings

no comment

Additional comments

The author have addressed my concern.

·

Basic reporting

no comment

Experimental design

no comment

Validity of the findings

no comment

Additional comments

The authors have addressed all of my comments and concerns in the revised version. I have no additional comments.